# School Fruit and Vegetables Scheme: Characteristics of Its Implementation in the European Union from 2009/10 to 2016/17

**DOI:** 10.3390/nu14153069

**Published:** 2022-07-26

**Authors:** Iris Comino, Panmela Soares, María Asunción Martínez-Milán, Pablo Caballero, María Carmen Davó-Blanes

**Affiliations:** 1Department of Community Nursing, Preventive Medicine and Public Health and History of Science, University of Alicante, 03690 Alicante, Spain; iriscomino@gmail.com (I.C.); panmela.soares@ua.es (P.S.); mariasuncion.m.m@gmail.com (M.A.M.-M.); mdavo@ua.es (M.C.D.-B.); 2Public Health Research Group, Department of Community Nursing, Preventive Medicine and Public Health and History of Science, University of Alicante, 03690 Alicante, Spain

**Keywords:** schoolchildren, fruit and vegetables, community nutrition, nutrition interventions, health promotion

## Abstract

The “School Fruit and Vegetables Scheme” (SFVS) was proposed in 2009/10 as a strategy to support the consumption of Fruit and Vegetables (FV), decrease rates of obesity, improve agricultural income, stabilize markets, and ensure the current and future supply of these foods. However, there is little information about how it was carried out in the EU. Given the potential of the SFVS to support healthier, more sustainable food systems, the objective of this study was to identify the characteristics of SFVS implementation from 2009/10 to 2016/17 in the EU. A longitudinal, observational, and retrospective study was carried out based on secondary data. A total of 186 annual reports of the Member States (MS) participating in the SFVS from 2009/10 to 2016/17 were consulted: European and national budget, funds used from the EU, participating schools and students, duration of the SFVS, FV offered, and application of sustainability criteria, expenditure per student, days of the week, the quantity of FV offered per student and other indicators were calculated. The majority of MS participated in the SFVS during the study period with a heterogeneous implementation pattern in terms of funds used, coverage, duration, quantity (totals and by portion), and cost of FV distributed per student. The sustainability criteria for the FV distribution were also not applied uniformly in all the MS. Establishing minimum recommendations for SFVS implementation are recommended to maximize the benefits of the SFVS. The results may be useful for planning new strategies to help address and improve current health and environmental problems.

## 1. Introduction

The low consumption of fruit and vegetables (FV) is an avoidable risk factor that contributes to an increase in non-transmissible diseases [1,2]. In 2017, 3.9 million deaths around the world were attributed to inadequate intake of FV [3]. Low consumption of FV in the European Union (EU) continues to be a problem related to social, physical, and economic accessibility [4]. Despite the efforts of governments to promote FV consumption, surveys of the child population indicate that average consumption is below recommended levels [5,6,7,8,9].

Considering that dietary habits established in childhood and adolescence tend to extend through adult ages [10,11], supporting the consumption of FV in children is a public health priority [6]. Diverse strategies have been carried out in school environments with this aim [12,13,14]. Available evidence suggests that school feeding programs are effective in improving the consumption of FV [15,16,17], specifically, those that promote the availability and accessibility of FV via distribution in schools [18,19,20,21,22,23,24].

Under the Common Agricultural Policy (CAP), the European Commission (EC) proposed increasing financial funding to support the consumption of FV in the school population [25]. The “School Fruit and Vegetables Scheme” (SFVS) was proposed in 2009/10 as a strategy to support the consumption of FV, decrease rates of obesity, improve agricultural income, stabilize markets, and ensure the current and future supply of these foods [26]. The SFVS consisted of European Union (EU) subsidized distribution of FV in pre-school centers, primary and secondary schools, and outside of school canteen hours. Furthermore, the SFVS included educational measures (EM), whose objective was to raise awareness among school children, connecting them not only to their health and well-being but also to agriculture, food production, and environmental sustainability issues. Along the lines of the Sustainable Development Goals (SDG), the SFVS recommended that distributed foods be organic, seasonal, and from local producers originating in the European Community. This is especially important considering the effect of the production of these foods on the food system [27].

From its application in 2009/10 until 2016/17 (the year before the implementation of the EU School Fruit, Vegetables and Milk Scheme), the SFVS benefitted from a budget of approximately 937,226,513.40 Euros. Member States (MS) interested in participating in the SFVS must develop an annual pre-strategy at the national or regional level. The overall target group of the scheme (from kindergartens to secondary schools) is set by the EU and MS choose their target group from this age range. In addition, MS must provide national funds to accompany EU support for the purchase and distribution of FV, the implementation of EM, and the monitoring and evaluation of the SFVS [26]. Although the SFVS was implemented in 2009/10, there is little information about how it was carried out in the EU. Given the potential of the SFVS to support healthier, more sustainable food systems, the objective of this study was to identify the characteristics of SFVS implementation from 2009/10 to 2016/17 in the EU.

## 2. Materials and Methods

A longitudinal, observational, retrospective study based on public secondary data was conducted and therefore no ethical approval was required. During the study period, between 21 and 25 of the 28 EU Member States (before Brexit in 2020) participated in the SFVS: Austria (AUT), Belgium (BEL), Bulgaria (BRG), Croatia (HRV), Cyprus (CYP), Czech Republic (CZE), Denmark (DNK), Estonia (EST), France (FRA), Germany (DEU), Greece (GRC), Hungary (HUN), Ireland (IRL), Italy (ITA), Latvia (LVA), Lithuania (LTU), Luxembourg (LUX), Malta (MLT), Poland (POL), Portugal (PRT), Romania (ROU), Slovakia (SVK), Slovenia (SVN), Spain (ESP), The Netherlands (NLD). Between 2017 and 2018, the annual reports of the 25 MS participating in the SFVS from 2009/10 to 2016/17 were obtained from the EC website, i.e., 186 reports in total [26].

With the information contained in the reports, by school year and MS, the database was constructed including the following variables: European and national budget (€), funds used from the EU (%), participating schools (n, %), participating students (n, %), duration of the SFVS (weeks, days of the week, days of FV deliveries), the quantity of FV offered (t, portions per participant), and application of sustainability criteria (local, seasonal, organic and community origin (%)). In addition, the following variables were calculated: expenditure per student (€), days of the week on which the SFVS was implemented, the quantity of FV offered per student (in kg), average portions offered per student, and the average price per portion. To calculate average participation (both for schools and for students), values were calculated for each MS during the middle of the period for the 2013 school year. To calculate the percentage of countries that applied the recommendation on local foods, seasonal foods, organic foods, and community-origin foods, the MS that provided data in their reports were used. To calculate the expenditure per student, the budget for each MS was divided between the students that complied with the SFVS in each country. Some data for these variables were missing but could be calculated from other data available in the reports. In no case have estimation methods been used for missing data. All indicators have been calculated with the available data (see Appendix A).

For the analysis, the described variables were organized into four sections: two for the total data of the EU, and another two with the data from the different MS, organized into eight sub-sections.

Sections for the study of SFVS implementation for the European Union globally:

General Characteristics of the SFVS by School Year for the EU As a Whole: Includes MS participants, budget (%), expenditure per student (€), % participation (schools and students), and duration (weeks and days of FV deliveries).

Quantity of FV Included in the SFVS by School Year and for the EU As a Whole: Includes the quantity of FV purchased/distributed (tons), FV per student (kg), average portions per student (n), weight per portion (g) and price per portion (€).

Sections and sub-sections for the study of the implementation of the SFVS for participating MS:

Involvement of the MS in Carrying out the SFVS: Included four sub-sections:

Economic investment in the SFVS by MS during the study period: Included EU funds used by each MS (%) and expenditure per student (€).

SFVS coverage by MS during the study period: Percentage of students covered by FV (%) and percentage of students covered by EM (%).

Days of FV deliveries and quantity of FV per student (kg) by MS during the study period.

Economic investment in terms of the coverage, duration, and quantity of FV distributed by the SFVS.

Characteristics of the Distribution of FV Among Students: Included four sub-sections:

Frequency of FV deliveries: Duration of the SFVS in weeks and days of FV deliveries

Quantities of FV distributed and percentage of students covered by duration. Relationships between FV per student (kg), duration (days of FV deliveries), and percentage of students covered (%).

Portions distributed per student. Relationships between price (€), weight (g), and the number of portions (n).

Sustainability criteria applied in the distribution of FV: included local, seasonal, organic and community origin criteria contemplated in the different country strategies. The values of the variables were calculated considering the number of times that countries mentioned these criteria in the strategies for each school year.

Sub-sections economic investment in the SFVS, SFVS coverage, days of FV deliveries and sustainability criteria were represented using maps, and the rest were represented using a scatter plot in which each point represents the mean value of a country during the study period. To assess the relationship between variables, we calculated Pearson’s correlation and linear regression. In addition, mean values were calculated for all variables. The statistical software package R [28] was used for the analysis and the graphs. The international abbreviation ISO-3166-1 ALPHA-3 was used for the nomenclature of the MS.

## 3. Results

### 3.1. General Characteristics of the SFVS by School Year in the Whole of the EU

Table 1 shows the general characteristics of the SFVS across the European Union from 2009/10 to 2016/17. During the eight years of implementation, between 21 and 25 MS participated. The percentage of EU funds used by the MS for the execution of the SFVS was irregular throughout the period, at 63% of the mean. The average participation in the SFVS—both for schools that ascribed to the SFVS and for students—increased progressively from the first to the final year, reaching a maximum in 2015/16 (44% of schools and 47% of students). The average days of FV deliveries by school year ranged from 44 to 61 days, except during the initial year in which it lasted 22 days. The average duration in weeks ranged from 13 weeks in the first year to 27 weeks in the following three years. The average over the whole period was 22 weeks. The Budget was correlated with Student expenditure (PC 0.874 *p*-value = 0.001), participation of Schools (PC 0.799 *p*-value = 0.007) and Students (PC 0.800 *p*-value = 0.006).

### 3.2. Quantity of Fruit and Vegetables Included in the SFVS by School Year in the Whole of the EU

Table 2 shows the quantity of FV distributed among students from 2009/10 to 2016/17 for all the countries participating in the SFVS. The quantity of FV distributed to participating students was not progressive, rather there were variations in the study period, reaching a maximum in the final school year (2289.20 tons). On average, each student who participated in the SFVS received between 2 and 5 kg of FV throughout the study period, distributed in portions of 122 g. The weight of each portion ranged from 50 g (in 2009/10) to 145 g (in 2013/14), and prices changed throughout the study period, shifting from an initial cost of 0.08 € in 2009/10, and 0.42 € during the following school year to a continuous decrease, reaching 0.29 € during the 2014/15 school year.

### 3.3. Involvement of the MS in Carrying Out the SFVS

#### 3.3.1. Economic Investment in the SFVS by MS during the Study Period

Figure 1 shows the percentage of EU funds used for the implementation by each MS and the expenditure per student. Four of the MS (SVK, MLT, HUN, and CZE) used 100% of EU funds to carry out the SFVS, while two (FRA and PRT) used just 25% of these funds during the period. There was important variation in the Netherlands and Austria.

In terms of expenditure per student, three countries (ITA, IRL, DNK) invested more than 25 € per student, although two of them (IRL, DNK) showed important variation during the period. In contrast, at least five countries (ROU, SVN, LUX, LVA, PRT) spent less than 6.5 € per student. Estonia stands out for its great variation during the period.

#### 3.3.2. SFVS Coverage by MS during the Study Period

Figure 2 shows the percentage of students covered by FV distribution and EM in each MS. Two countries (FRA, DNK) barely reached 10% of students, while five (HRV, HUN, CZE, LVA, and MLT) reached between 80% and 100% of students during the period. Five other countries (ROU, CYP, AUT, BGR, LTU) stand out for the great variation in the number of students reached (between 25% and 100%) throughout the study period.

In terms of the percentage of students covered by EM, 7 of the 25 MS (BGR, DEU, GRC, ITA, LVA, LUX, NLD) were able to cover all the students registered in the SFVS with EM in all the editions. However, three countries (DNK, AUT, BEL) did not include them and did not provide data. In the rest of the MS, the percentage of students covered ranged from 15% to 90%, with great variation in most cases.

#### 3.3.3. Days of FV Deliveries and Quantity of FV per Student (kg) by MS during the Study Period

Figure 3 shows the days of FV deliveries and quantity of FV per student (kg) in each MS. Three countries (DEU, ROU, DNK) presented a variation of 75 to 200 days during the period, while the rest did not reach 50 days. Three countries (AUT, CZE, GRC) did not report sufficient data for this variable.

The quantity of FV distributed per student varied between 2.5 and 20 kg in the countries. Six MS (PRT, IRL, SVK, BEL, ESP, and MLT) distributed around 2.5 kg of FV during the whole period, while four (BGR, HUN, POL, LTU) distributed between 7 and 15 kg per student. Denmark reached a level of between 15 and 20 kg throughout the period.

#### 3.3.4. Economic Investment in Terms of the Coverage, Duration, and Quantity of FV Distributed by the SFVS

Figure 4 represents the relationship between the percentage of EU funds used by the MS and the percentage of students covered by the SFVS (Pearson correlation 0.637, *p*-value < 0.001). As the percentage of EU funds used increases, the percentage of students covered by the SFVS also increases. The MS that used between 75% and 100% of EU funds reached a high level of coverage (above 75%), but they distributed lower levels of FV. The MS located below the regression line did not reach the percentage of students corresponding to the use of EU funds.

Figure 5 shows the relationship between the days of FV deliveries, the percentage of EU funds used by the MS (Pearson correlation −0.440, *p*-value = 0.041), and the expenditure per student (€). It can be observed that, in most cases that used more than 75% of the budget, the days of FV deliveries did not reach 50, and the expenditure per student was irregular.

Regarding the relationship between the quantity of FV distributed and the expenditure per student (Pearson Correlation 0.675, *p*-value < 0.001) and duration (Figure 6), it can be observed that most MS distributed between 4 and 8 kg of FV with an expenditure of up to 20 euros per student, and highly variable duration.

Figure 7 shows the relationship between expenditure per student (€), percent of students covered (Pearson Correlation 0.416, *p*-value = 0.043), and days of FV deliveries. It can be observed that the majority of participating MS maintained a similar expenditure per student (of approximately 10 €); however, the percentage of students covered by EM and the days of FV deliveries were more variable. A large group of MS managed to cover between 75% and 100% of students with EM, but there was a heterogeneous duration. Belgium and Austria did not include EM or did not report data in this area.

### 3.4. Characteristics of FV Distribution among the Students

#### 3.4.1. Frequency of FV Deliveries

Figure 8 shows the frequency of FV deliveries in each MS accounting SFVS duration in weeks and days of FV deliveries. Most MS designed schemes of between 2 and 3 days per week, with differences in the number of weeks of duration during the period (from 15 weeks to 35 weeks, approximately). Ireland stands out for making deliveries during one week of the school year, and Romania and Denmark stand out for distributing FV five days per week for approximately 25 weeks.

#### 3.4.2. Quantities of FV Distributed and Percentage of Students Covered by Duration

Figure 9 shows the relationship between days of FV deliveries, quantity per student, and the percentage of students covered by the SFVS. It can be observed that the majority of MS implemented the scheme for between 25 and 50 days, with a distribution of less than 10 kg per student. MS located below the line distributed less FV than what would correspond to them, according to the programmed duration. Ireland stood out for distributing less FV for fewer days and for serving less than 25% of students during the period.

#### 3.4.3. Portions Distributed per Student

Figure 10 shows the linear relationship between price and weight of portions. The size of the dots represents the number of portions supplied per student. It can be observed that as weight increases, the cost increases proportionally. However, all of the countries, except five, supplied portions of insufficient weight concerning cost. Except for Denmark, none of the countries surpassed 100 portions per student, and the average portion weight was less than 125 g.

#### 3.4.4. Sustainability Criteria Applied in FV Distribution

Figure 11 represents the application of sustainability criteria (local, seasonal, organic, and sourcing from the European Community) for the FV distributed during the 2009/10–2016/17 period, by participating MS. In general, the most applied criteria related to seasonal produce, and organic production was the least applied criteria. Three countries (ESP, FRA, DEU) applied the four criteria for the whole period, while two (DNK and ROU) did not apply any criteria.

## 4. Discussion

This study identified characteristics of the implementation of the School Fruit and Vegetables Scheme from 2009/10 to 2016/17 in the European Union. The majority of MS participated in the SFVS during the study period, with a heterogeneous implementation pattern in terms of funds used, coverage, duration, quantity (totals and by portion), and cost of FV distributed per student. This heterogeneity shows different involvement of the MS that implemented the SFVS. Few countries used the entire amount of EU funds destined for SFVS implementation, made sufficient investment per student, or reached total coverage, both in terms of FV provision as well as EM. In general, the days of FV deliveries did not reach 50, and quantities of FV distributed were less than 5 kg per student. The distribution of FV also showed different characteristics between the MS. The range of periodicities of FV distributions to schools was wide, with a variation among countries of between 1 and 5 days in periods of between 1 and 35 weeks. In general, the average portion weight provided to schools was less than 125 g per day, and no country surpassed 100 portions. Temporality was the sustainability criteria most applied in the distribution of FV by different MS.

In all of the EU, the participation of schools and students in the SFVS increased across the period, however, the heterogeneity of EU funds utilized by MS had implications for the coverage reached both in terms of FV distribution as well as EM. In general, in all of the EU during the study period, the funds destined for the implementation of the SFVS seem to have been insufficient to cover the total number of students. The MS that most approximated maximum coverage quotas did so supplying little FV. In fact, only four of twenty-five MS covered the total number of students with FV, and only seven included EM in all the editions. These results are similar to those identified in the USA for the Fresh Fruit and Vegetable Program (FFVP), where coverage of 100% of the students was not achieved [29]. There could also be disadvantages for students who were not covered by the SFVS, given their exclusion from potential benefits. On one hand, there is evidence that the availability of FV in schools promotes their consumption [30,31]. On the other hand, interventions that include EM along with FV distribution are more effective in promoting the consumption of these foods in schools [32,33,34]. Furthermore, the educational measures contemplated in the SFVS could contribute to students’ knowledge of individual health and environmental benefits related to FV consumption [35,36]. However, it is important to note that implementation of these measures falls on teachers, thus it is dependent on their willingness and perception of the additional effort involved. Providing training and support could facilitate the implementation of these measures in schools [37,38].

Despite the percentage of EU funds used permitting greater coverage of students in those MS that made a higher investment in the development of the SFVS, in general, it did not contribute to the duration nor the expenditure per student. In most cases using more than 75 percent of EU funds, the days of FV deliveries did not reach 50 days. One of the most commonly tested techniques for increasing FV consumption is repeated exposure [39]. An estimated 59 and 66 days are needed for successful habit formation [40,41], which is greater than the average number of days of provision of FV to students in the SFVS evaluated. While some countries with an expenditure per student less than the average and a long-duration plan distributed greater quantities of FV than other countries, only Denmark surpassed 15 k and had a long-duration, with significant expenditure, but with a limited reach in terms of the total number of students.

Nor did students covered by the SFVS receive enough FV constantly. As shown by the results, the quantities distributed had variation over time, providing to each school (with some exceptions) less than 5 kg per school year. These data are similar to the findings in the Polish evaluation [38]. It is known that the direct provision of FV in the school setting can have success in increasing FV consumption [10,22]. For this reason, there should be increased efforts to increase the quantity of FV distributed to students. While it is true that distribution can be influenced by different factors, plans and programs that are coordinated among different sectors could help to build alliances with all the involved parties that could contribute to achieving common objectives [42,43]. According to our results, the average portion weight was less than 125 g, and there was great variability between portion cost and portion weight among the countries. This suggests that some MS have difficulty providing the minimum portion weight of FV, which affects the consumption of the 400 g daily consumption recommended by the WHO [44,45]. Furthermore, the high heterogeneity by country in terms of FV deliveries, ranging from once per day during one week to 2.5 times per day over 35 weeks, could influence consumption habits at schools. In Germany, SFVS with FV deliveries three or two times a week led to a significant increase in FV intake [46]. For this reason, establishing a minimum quantity of FV per student and the periodicity of deliveries is recommended.

According to the SDG, it would be helpful if FV distributed in schools could contribute to reducing the environmental footprint, however, sustainability criteria were not applied uniformly in all the MS. While some countries distributed local, seasonal, and organic foods or sourced from the European Community during the study period, others did not do so in any of the editions of the SFVS. It should be noted that economic policies have promoted the importation of cheap FV to the detriment of varieties produced locally [4]. Given the limited budget, the scheme may not have a significant direct impact on market balance. Additionally, the distribution of funds to MS should be accompanied by sustainability commitments to favor uniformity of the characteristics of FV distribution in different countries. Stronger ties are needed among the different agents involved in the whole food system to integrate sustainable practices in terms of production, harvest, processing, and consumption [4].

Limitations: When interpreting the results, it should be taken into account that the data used are reported by the participating MS themselves, which could be considered a limitation of the study. However, the information reported by MS was uniform, which permitted establishing comparisons in terms of SFVS implementation in all of them. Moreover, we can measure the success of the SFVS in terms of participation, duration, or FV distributed but not in terms of the evolution of overweight and obesity, FV intake, or knowledge acquired with the educational measures because this information is not available.

## 5. Conclusions

Our results suggest that the implementation of the SFVS in MS has been very heterogeneous, which means that EU students do not benefit equally from the SFVS. The purchase of fruit and vegetables from local producers for distribution in schools could have a positive impact on agricultural production and also on the consumption of these products among the school population, as shown in prior studies. However, the lack of continuity in the execution, as well as the low number of days of FV deliveries, could limit its potential benefits. Establishing minimum recommendations for SFVS implementation, including the number of days, percentage of students covered, and quantities of FV distributed, are recommended to maximize the benefits of the new School Fruit, Vegetables and Milk Scheme.

## Figures and Tables

**Figure 1 nutrients-14-03069-f001:**
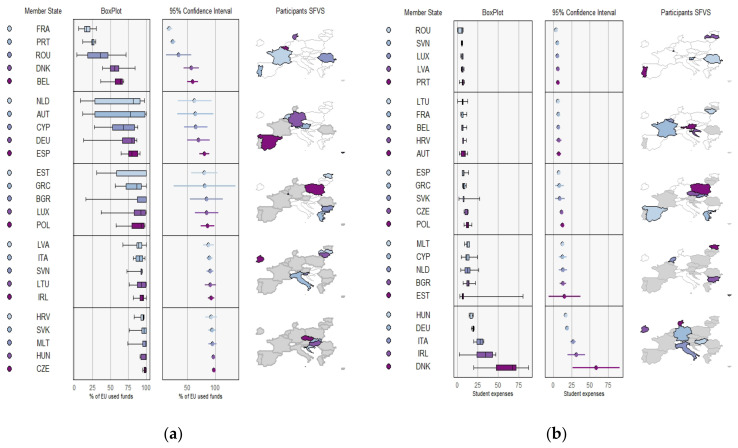
(**a**) Percentage of EU funds used by each MS; (**b**) expenditure per student in each MS.

**Figure 2 nutrients-14-03069-f002:**
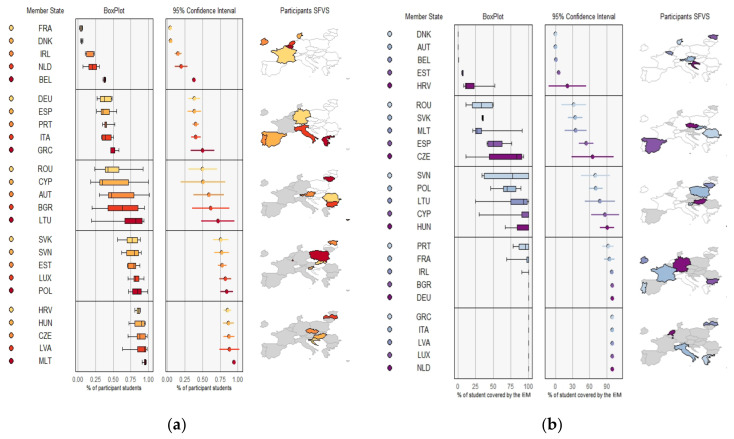
(**a**) Percentage of students covered by FV distribution; (**b**) Percentage of students covered by EM.

**Figure 3 nutrients-14-03069-f003:**
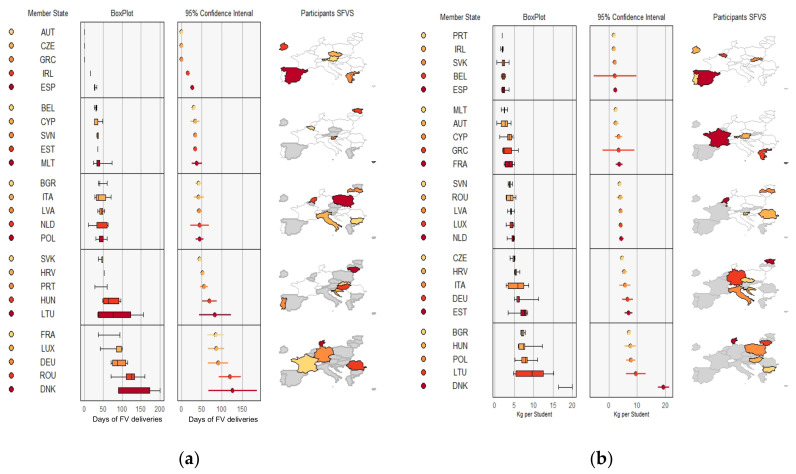
(**a**) Days of FV deliveries; (**b**) quantity of FV per student in each MS (kg).

**Figure 4 nutrients-14-03069-f004:**
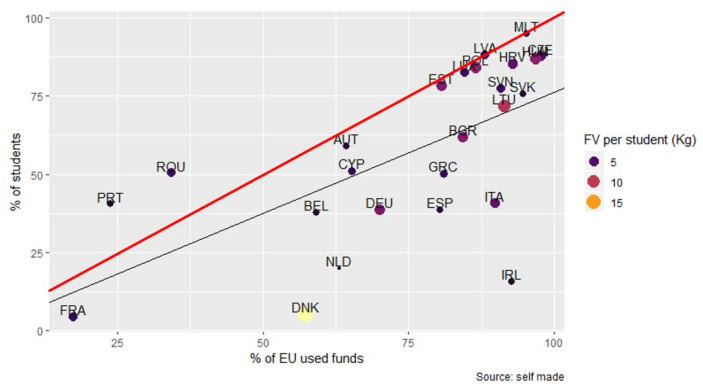
Relationship between the EU funds used (%), students covered (%), and the quantity of FV distributed (kg) by the SFVS. Regression line (black): represents the percentage of student beneficiaries of the SFVS using 100% of EU funds. Identity function (red): % of EU funds used by MS = % of students covered. The size of the dot represents the quantity of FV distributed per student (kg).

**Figure 5 nutrients-14-03069-f005:**
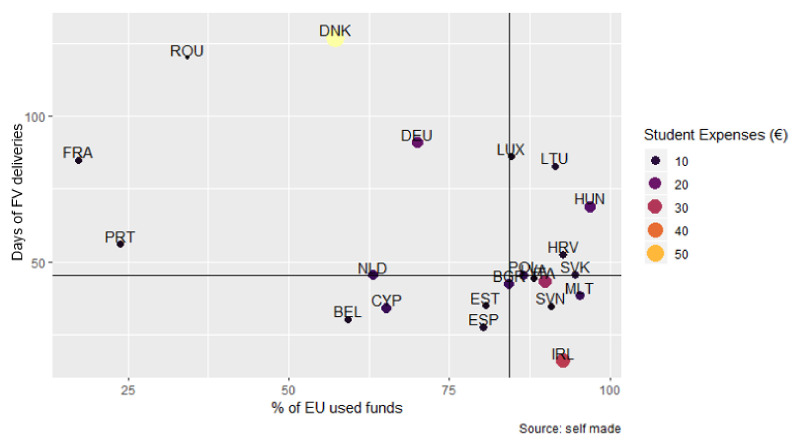
Relationship between the days of FV deliveries (days), the percentage of EU funds used by the MS, and the expenditure per student (€) of the SFVS. Horizontal line: represents the average of days of FV deliveries in each MS; Vertical line: represents the average % of EU funds used by the MS.

**Figure 6 nutrients-14-03069-f006:**
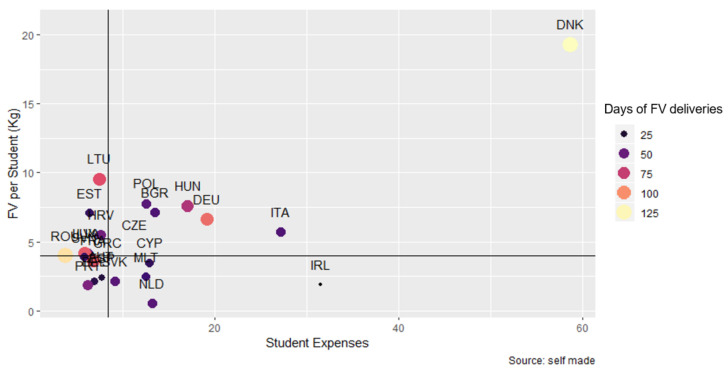
Relationship between fruit and vegetables distributed per student (kg), expenditure per student (€), and days of FV deliveries. Horizontal design: median FV distributed per student (3.98 kg); Vertical line: median expenditure per student (8.38 €).

**Figure 7 nutrients-14-03069-f007:**
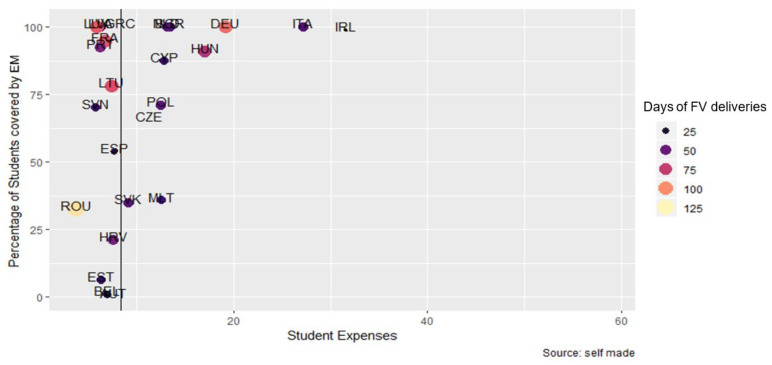
Relationship between the expenditure per student (€), the percentage of students covered by EM, and the days of FV deliveries. Vertical line: average expenditure per student (8.38 €).

**Figure 8 nutrients-14-03069-f008:**
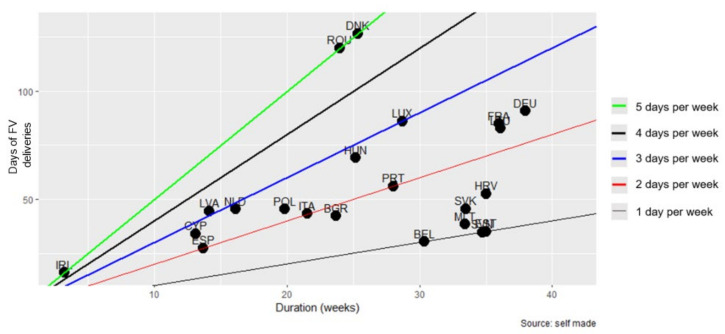
Relationship between SFVS duration in weeks and days of FV deliveries.

**Figure 9 nutrients-14-03069-f009:**
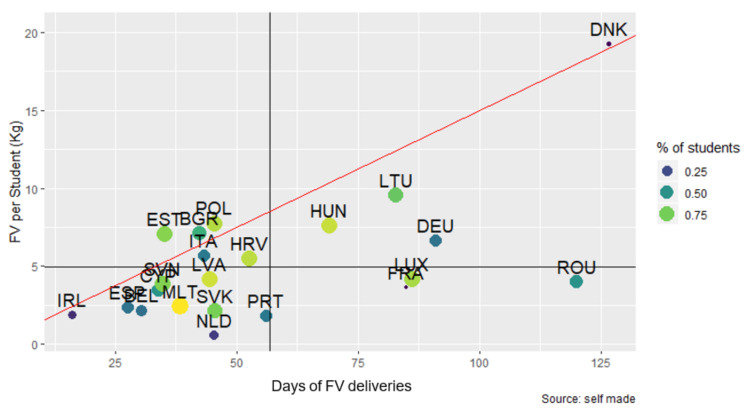
Relationship between days of FV deliveries, the quantity of FV per student, and the percentage of students covered by the SFVS. Horizontal line: average quantity of FV per student (4.95 kg); Vertical line: average days of FV deliveries (56.9 days). Red line: a linear equation that represents the supply of 150 g per day, based on the duration of the SFVS.

**Figure 10 nutrients-14-03069-f010:**
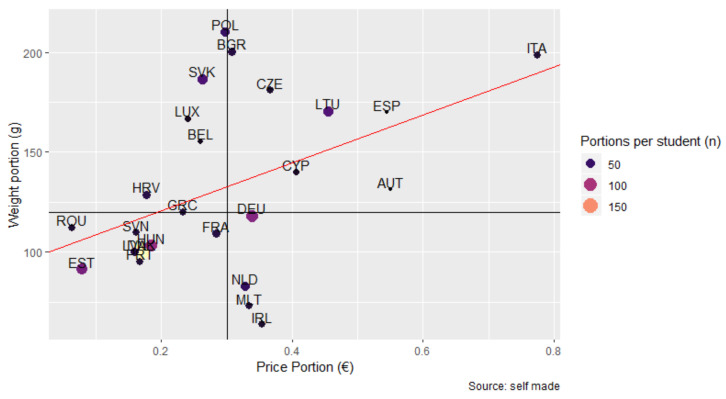
Relationship between portion price, portion weight, and the number of portions distributed per student by the SFVS. Horizontal line: minimum portion weight (120 g); Vertical line: average portion price (0.30 €). Regression line (red): represents the average portion price by weight.

**Figure 11 nutrients-14-03069-f011:**
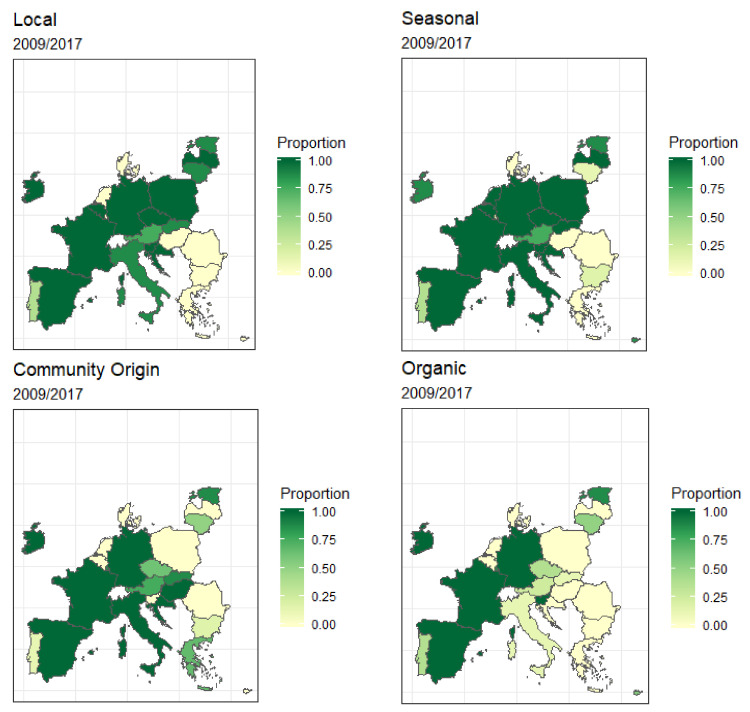
Sustainability criteria of the SFVS from 2009/10 to 2016/17. Average local criteria; Average temporality criteria; Average criteria European Community origin. Average organic criteria.

**Table 1 nutrients-14-03069-t001:** General characteristics of the EU School Fruit and Vegetables Scheme from 2009/10 to 2016/17.

School Year	MS ^1^	Non-Participating MS	Budget	Student Exp ^2^	Participation		Duration	
	n		%EU	€	%Schl ^3^	%Stud ^4^	Days of FV Deliveries	Week
09/10	21	BGR; HRV; GRC; LVA	33.11	9.35	18	21	21.75	12.96
10/11	24	HRV	61.91	14.11	38	38	59.38	26.67
11/12	23	HRV; GRC	61.77	12.48	33	34	60.61	26.63
12/13	23	HRV; CYP	71.71	15.46	39	38	54.93	26.99
13/14	25		72.98	13.42	35	42	44.37	17.75
14/15	24	GRC	66.07	12.87	28	29	44.25	16.92
15/16	24	GRC	69.25	13.38	44	47	49.05	20.38
16/17	22	GRC; LUX; ROU	67.93	13.00	44	45	57.59	25.59
Total	25		63.09	13.01	35	37	48.99	21.74

^1^ MS: Member State; BGR: Bulgaria; HRV: Croatia; GRC: Greece; LVA: Latvia; CYP: Cyprus; LUX: Luxembourg; ROU: Romania. ^2^ Student Exp: Student Expense. ^3^ %Schl: percentage of schools. ^4^ %Stud: percentage of students.

**Table 2 nutrients-14-03069-t002:** Quantity of Fruit and Vegetables included in the SFVS from 2009/10 to 2016/17.

School Year	Quantities of FV Purchased/Distributed	FV Per Student/School Year	Average Portions Offered per Student	Average Portion Weight	Average Portion Price
	Tons	kg	n	g	€
09/10	1707.9	1.95	15.57	49.35	0.08
10/11	631.1	4.88	51.76	139.02	0.42
11/12	1358.7	4.18	42.80	135.97	0.41
12/13	1083.1	5.54	54.23	139.03	0.40
13/14	1025.7	4.57	44.14	144.64	0.37
14/15	408.0	4.81	39.72	125.92	0.29
15/16	1743.0	3.89	41.23	120.63	0.31
16/17	2289.2	4.10	40.67	124.36	0.37
Average	1280.84	4.24	41.27	122.37	0.33

FV = fruit and vegetables.

## Data Availability

Data supporting reported results were obtained between 2017 and 2018 from the EC web page, using the following link: https://ec.europa.eu/agriculture/sfs_en (accessed on 28 December 2017). However, this link is not currently available because the scheme changed with the entry into force of Regulation (EU) 2017/39 which lays down the basis for the new School Fruit, Vegetables and Milk Scheme. For this reason, the authors add to the Appendix A an example of a report from a member state and a database with all available variables for all countries for the year 2013. In addition, the authors provide the complete database upon request to the corresponding author.

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
