# Peer review of "School Fruit and Vegetables Scheme: Characteristics of Its Implementation in the European Union from 2009/10 to 2016/17"

_nutrients, 2022, doi:10.3390/nu14153069_

Round 1
Reviewer 1 Report
Authors used the secondary data to identify the characteristics of SFVS implementation from 2009 to 2017 in the EU. The results show that most MS participated in the SFVS with a heterogeneous implementation pattern in terms of funds used, coverage, duration, quantity, and cost. The sustainability criteria for the FV distribution were not applied uniformly in all the MS. The findings might be helpful for policy development and public health strategies to improve healthy eating behaviors and health outcomes. The link you provided is applicable since 2017 till now. I am not sure how did you get the data; any approval or agreement is needed? In addition, the distribution of this scheme is illustrated clearly and interactively on the website related to all the indicators that you presented in your paper in different ways. I am afraid that this might not be qualified as an original research article. The following comments might be helpful for your manuscript.
Abstract
Please indicate the study design and settings, and specify the measures.
Introduction
Please list out all the member states where the SFVS was implemented.
Methods
As I understand this is a school scheme, not for the general public. Therefore, authors should specify the study settings in the methods.
In Line 86, which are variables that you organized into 4 categories? The word “categories” might not be appropriate to describe what you aim to measure.
What is the procedure for extracting and managing data? Which software did you use to analyze the data?
Have you received ethical approval for conducting this study?
Which software was used to analyze the data and draw the figure?
Results
It is useful to report the number of students in total and by the state who were included and analyzed in this study.
Any trending analysis?
In Table 1, what is the meaning of “No. included”? while you provide the name of some member states.
In Table 2, what is the unit of FV consumption by students? Is that per week or per month, or per year or the whole study period? A figure is preferred to view the trend.
In Figure 4, could you test the association strength between the percentage of EU funds (not UE) by the MS and the percentage of students covered by the SFVS. Pearson or Spearman correlation is suggested.
In Figure 5, Figure 6, and Figure 7, it’s difficult to see the trend/pattern of the relationships. There might be a better illustration and test to check these relationships.
Figure 8 does not provide any sense for the study purpose. I suggest deleting it.
Please provide the supplementary table for each figure.
Discussion
I can’t get what you mean in the limitation section. Please readdress this part.
Finally, English writing should be intensively improved throughout the manuscript, in terms of vocabulary used and grammar.
Author Response
Reviewer 1
Response: We appreciated your time in reviewing this work. In this new version, we have incorporated the suggestions of the reviewer, and we believe that it has contributed to improvements in the manuscript.
- Authors used the secondary data to identify the characteristics of SFVS implementation from 2009 to 2017 in the EU. The results show that most MS participated in the SFVS with a heterogeneous implementation pattern in terms of funds used, coverage, duration, quantity, and cost. The sustainability criteria for the FV distribution were not applied uniformly in all the MS. The findings might be helpful for policy development and public health strategies to improve healthy eating behaviors and health outcomes. The link you provided is applicable since 2017 till now. I am not sure how did you get the data; any approval or agreement is needed? In addition, the distribution of this scheme is illustrated clearly and interactively on the website related to all the indicators that you presented in your paper in different ways. I am afraid that this might not be qualified as an original research article. The following comments might be helpful for your manuscript.
Response: Thank you very much for this comment. The contribution of this comment has undoubtedly improved the methodology section of the manuscript.
From the information we had reflected in the text, it was not clear how original and effortful the reporting and database creation was. We have synthesised more than 175 reports and these reports were public, so it was not necessary to ask for any permission or approval. However, as you comment, the link from which we obtained the information is not currently available because the scheme changed with the entry into force of Regulation (EU) 2017/39 laying down the basis for the new School Fruit, Vegetables and Milk Scheme. Therefore, information can be found from 2017 to 2021 (even 2022 in some MS) related to this new scheme, but not to the previous one. We have clarified in the text these considerations. Please, see the first paragraph of the methods section.
The database was created before the removal of these data. Indeed, one of the strengths of this study is that it makes available to researchers the data contained in the reports that are no longer available on the European Commission's website. Thus, we have modified the "Data Availability Statement" section at the end of the document and indicated that the database compiled from the more than 175 reports is available to the scientific community upon request to the authors. Please, see “Data Availability Statement”.
Thank you again for this valuable comment.
- Abstract. Please indicate the study design and settings, and specify the measures.
Response: We have included this information in the abstract. Please see the abstract, lines 14-15.
- Introduction. Please list out all the member states where the SFVS was implemented.
Response: Thank you for this suggestion. We have mentioned each member state but in the methods section instead of the introduction section. Please, see lines 69-75.
- Methods.
- As I understand this is a school scheme, not for the general public. Therefore, authors should specify the study settings in the methods.
Response: In response to the comment, we have included this information in the introduction part. Please see lines 60-62
- In Line 86, which are variables that you organized into 4 categories? The word “categories” might not be appropriate to describe what you aim to measure.
Response: Thanks for the comment, we believe that the word "section" is more appropriate than the word "categories". We have made this correction in the text. Please, see methods part.
- What is the procedure for extracting and managing data? Which software did you use to analyze the data?
Response: Thank you very much for this clarification. This answer is related to the first comment. The data were extracted from the evaluation reports that each MS submit each year at the end of the school year, i.e. we handle a total of 170 reports. This information has been included in the first paragraph of the methods.
As for the software, we used R to analyse the data. This information is described in lines 136-137.
- Have you received ethical approval for conducting this study?
Response: Since the research does not involve human subjects, human material, human tissues or human data, ethical approval is not required for the conduct of the study. We have included a sentence in the text to clarify this information. Please, see line 71 in the methods part.
- Which software was used to analyze the data and draw the figure?
Response: The software that we used to analyze the data, was R. This information is described in lines 133-137.
- Results.
- It is useful to report the number of students in total and by the state who were included and analyzed in this study. Any trending analysis?
Response: Differences in the size of the populations in the different MS make comparison between countries difficult. However, we understand the reviewer's concern, as having the primary data makes the reproducibility of the study more feasible. For this reason, we have decided to provide the database compiled from the reports so that researchers can reproduce the study. We have reflected this information in the "Data availability statement" at the end of the document.
Regarding trend studies, we have only analysed the trend for EU aggregate data. A trend analysis of all participating MS is proposed as a future study because of the amount of data it contains.
- In Table 1. what is the meaning of “No. included”? while you provide the name of some member states.
Response: Thank you for this appreciation. We consider that it is necessary to change the text. We have written “Non-participating” instead of “No included”. Please, see Table 1.
- In Table 2, what is the unit of FV consumption by students? Is that per week or per month, or per year or the whole study period? A figure is preferred to view the trend.
Response: Thank you for the comment. The variable FV distributed per student refers to the kg of FV distributed per student by school year. We have changed the name of the column to clarify it.
As regards the proposed figure for the trend, we agree. However, we have created it and it is not legible for the reader when trying to represent the 5 variables. So, we have preferred to keep the table as it captures the original data more accurately.
- In Figure 4, could you test the association strength between the percentage of EU funds (not UE) by the MS and the percentage of students covered by the SFVS. Pearson or Spearman correlation is suggested. In Figure 5, Figure 6 and Figure 7, it’s difficult to see the trend/pattern of the relationships. There might be a better illustration and test to check relationships.
Response: We greatly appreciate these comments. We have revised the text and decided to add Pearson's correlation for each figure.
Reviewer 2 Report
Thank the editorial board for the opportunity to review the work by Comino et al. The authors identify the characteristics of SFVS implementation from 2009 to 2017 in the EU. The results is useful for planning new strategies to help address and improve current health and environmental problems.
See below a few comments.
1. Line 129: I think the percentage of EU funds used by the MS has also increased dramatically, by 51%, from the first to the final year, and its pattern of change is similar to that of the average participation for the percentage of students.
2. Line 132: If 44.3 days can be considered 45 days, then 60.6 days should also be considered 61 days.
3. Line 133-134: I don't understand why the " remained constant " is used. What does this sentence mean?
4. Please being consistent with the expression of year throughout the paper, for example, “from 2009/10 to 2016/17” in the title of Table 1 and “from 2009 to 2017” in the title of Table 2.
5. What is the full name of UE?( e.g. table 1, Line 190, Line 193 ) I think you want to express the abbreviation of European Union, EU. Please check.
6. Table 1:
• The full stop should be removed from the “Student Exp2.”
• The last line represents the average value of each indicator, so I think the “total” is inappropriate.
• 0.35 and 0.37 should be revised as 35 and 37 respectively.
7. Please, be consistent with the number of decimals included (1 or 2?) in Table 1 and Table 2.
8. Line 158: Is it possible to use the abbreviations of the countries in the figure here so that the readers can easily find the corresponding results in the figure?
9. Line 180: For consistency, the unit of the quantity of FV per student should be placed after “per student”.
10. Any chance to introduce the data missing of this survey and the main treatment methods of the missing data.
Author Response
Reviewer 2
Thank the editorial board for the opportunity to review the work by Comino et al. The authors identify the characteristics of SFVS implementation from 2009 to 2017 in the EU. The results is useful for planning new strategies to help address and improve current health and environmental problems. See below a few comments.
Response: Thank you very much for your time and the comments you have provided. We believe they have been very helpful in improving the manuscript.
- Line 129: I think the percentage of EU funds used by the MS has also increased dramatically, by 51%, from the first to the final year, and its pattern of change is similar to that of the average participation for the percentage of students.
Response: Thank you very much. We have included this remark in the text. Please see the first paragraph of results.
- Line 132: If 44.3 days can be considered 45 days, then 60.6 days should also be considered 61 days.
Response: Thank you very much for this correction. We have modified the data. Please, see line 147.
- Line 133-134: I don't understand why the " remained constant " is used. What does this sentence mean?
Response: We appreciate this comment because the phrase may cause confusion. We have rewritten this part. Please see lines148-150
- Please being consistent with the expression of year throughout the paper, for example, “from 2009/10 to 2016/17” in the title of Table 1 and “from 2009 to 2017” in the title of Table 2.
Response: Thank you very much for this correction. We have unified this period throughout the document.
- What is the full name of UE? (e.g. table 1, Line 190, Line 193) I think you want to express the abbreviation of European Union, EU. Please check.
Response: Indeed, it is a typo. We have checked the whole document to change it.
- Table 1:
- The full stop should be removed from the “Student Exp2
- The last line represents the average value of each indicator, so I think the “total” is inappropriate.
- 35 and 0.37 should be revised as 35 and 37 respectively.
- Please, be consistent with the number of decimals included (1 or 2?) in Table 1 and Table 2.
Response: Thanks for the comments. We have incorporated the suggested changes in Table 1.
- Line 158: Is it possible to use the abbreviations of the countries in the figure here so that the readers can easily find the corresponding results in the figure?
Response: This is a good suggestion. We have done a test including this information, but it complicates the legibility of the figures. The figure captions are too long. However, we have added a list of all participating countries and their abbreviations in the methods section.
- Line 180: For consistency, the unit of the quantity of FV per student should be placed after “per student”.
Response: Thank you. We have incorporated this suggestion. Please, see line 197
- Any chance to introduce the data missing of this survey and the main treatment methods of the missing data.
Response: Thank you for your contribution. Some of the reports did not include some variables but could be calculated indirectly using other variables. In no case was the data estimated. It was preferred to leave it as missing data. We have added this information in the text. Please, see lines 93-96.
Reviewer 3 Report
The paper is interestingly written and deals with a topic of paramount importance in light of the obesity epidemic among children around the world.
I think the paper should be upgraded to recommend to the EU or member states concrete recommendations for the future.
My comments can be seen in the attached file.

Author Response
Reviewer 3
The paper is interestingly written and deals with a topic of paramount importance in light of the obesity epidemic among children around the world. I think the paper should be upgraded to recommend to the EU or member states concrete recommendations for the future. My comments can be seen in the attached file.
Response: We appreciated your time in reviewing this work. In this new version, we have incorporated the suggestions and we believe that it has contributed to improvements in the manuscript.
- It is important to state the reasons for this in different countries and cultures. In Israel, for example, there is no problem with the availability of fruits and vegetables, but the prices are very high.
Response: Thank you very much for this comment. We have extended the description of accessibility to FV in the same sense as the FAO does by stressing that the problem of low FV consumption is not availability. Please see line 32.
- What was the process of joining the project? Line 59-60
Response: In this new version, we include information related to the process of joining the scheme. Please see lines 59-60 and 62-64.
- Beyond identifying the characteristics, what is the applied value of the study? What insights do researchers seek to gain?
Response: Since a lot of money is allocated for the implementation of this scheme in the EU, we wanted to check how the budget is handled and what needs to be improved in the new School Fruit, Vegetable, and Milk Scheme. For example, as mentioned in the conclusions, the study highlights a large heterogeneity that should be corrected in order to ensure that all European children have access to the plan.
- A table is missing with details about each member state, how many schools operated under the program, their percentage of all schools in the country and how many students in total in each country were exposed to the program.
Response: A table with data on 25 member states for 8 years and each variable was unwieldy, so we synthesized this information with percentages in Table 1 (for both schools and students). Also, you can find information on this in Figure 2 (a). We provide a table as supplementary material.
- Please add practical recommendations on how to promote the project, especially in countries where it has performed less well.
Response: We included the following information in the conclusions: “Establishing minimum recommendations for SFVS implementation, including the number of days of execution, percentage of students covered, and quantities of FV distributed, are recommended in order to maximize the benefits of the SFVS”. But we don't know if you are referring to something more specific. If so, we would be happy to provide the information you request.
- What are the insights for continuing the activities?
Response: With the entry into force of Regulation (EU) 2017/39, the scheme was modified to distribute in addition to fruit and vegetables, milk in schools. The idea remains the same, but a new food group has been incorporated, which we are currently evaluating. If you have any suggestions, we will be happy to incorporate them.
- Was the project successful and met the goals to continue?
Response: Thank you for your comment. As mentioned in the text, we can partially evaluate success through the percentage of students benefiting from the programme, the quantity of FV distributed per student or the duration. But there are other indicators such as the evolution of overweight and obesity, the intake of FV or the knowledge acquired in the educational activities that could not be measured due to lack of data. We have expanded the limitations paragraph with these aspects. Please, see limitations.
Round 2
Reviewer 1 Report
Dear Authors,
Thanks for the great effort to improve the manuscript. I understand that this can provide a picture of SFVS before 2017 to the readers.
Since the data were collected online via the link that you provided but it is unavailable now. You also mentioned that the data were obtained from more than 175 reports which were public but they are also not available to the public now. Therefore, I strongly suggest providing the compiled datasets from more than 175 reports in the supplementary materials.
In addition, in your responses, you mentioned that "... we handle a total of 170 reports. This information has been included in the first paragraph of the methods. ...", but in the methods, you wrote more than 175 reports. Which is the correct number? Please state the exact number of the reports. The word "more than" is not a scientifically written way to describe the size of data.
Best,
Author Response
Dear Authors,
Thanks for the great effort to improve the manuscript. I understand that this can provide a picture of SFVS before 2017 to the readers.
Since the data were collected online via the link that you provided but it is unavailable now. You also mentioned that the data were obtained from more than 175 reports which were public but they are also not available to the public now. Therefore, I strongly suggest providing the compiled datasets from more than 175 reports in the supplementary materials.
In addition, in your responses, you mentioned that "... we handle a total of 170 reports. This information has been included in the first paragraph of the methods. ...", but in the methods, you wrote more than 175 reports. Which is the correct number? Please state the exact number of the reports. The word "more than" is not a scientifically written way to describe the size of data
Dear Reviewer,
Thank you very much for your dedication to this manuscript.
We understand your concern about the source of the data. Indeed, we share your concern. As we mentioned in the manuscript, the evaluation reports for each school year were available at https://ec.europa.eu/agriculture/sfs_en. With this link, you had access to each participating country and to each evaluation reports by school year. These documents were used to build the database in 2017. After changing the interface of the website, these reports disappeared from the website of the European Commission. We were also quite surprised when European Commission withdrew of the documents from their webpage. Unfortunately, we did not archive them due to the large number of documents and because we felt it was not necessary. Therefore, we cannot attach you all the files. We have only the reports for Spain because, previously, we conducted an in-depth study on this country:
https://www.mdpi.com/1660-4601/16/20/3898
Therefore, we consider our paper is valuable because it provides information that is no longer available.
In any case, you are right, the number of documents consulted was very unspecific. The number was 186; one for each edition in which the MS participated, therefore, It is the sum of the participating MS shown in column 2 of Table 1. We have edited the document.
If you have any other suggestions, please let us know. We are glad to answer your questions. Your suggestions have really helped us to improve our manuscript.
Thank you very much,
Pablo Caballero.
Reviewer 3 Report
The corrections made by the authors shed light on several aspects that were not clear in the initial paper. I recommend publishing the present version.
Author Response
Thank you for your inputs and suggestions.